# Off-Policy Evaluation via the Regularized Lagrangian

**Mengjiao Yang**[*1]   **Ofir Nachum**[*1]   **Bo Dai**[*1]   **Lihong Li**[1]   **Dale Schuurmans**[1,2]

[1]Google Research    [2]University of Alberta

## Abstract

The recently proposed *distribution correction estimation* (DICE) family of estimators has advanced the state of the art in off-policy evaluation from behavior-agnostic data. While these estimators all perform some form of stationary distribution correction, they arise from different derivations and objective functions. In this paper, we unify these estimators as regularized Lagrangians of the same linear program. The unification allows us to expand the space of DICE estimators to new alternatives that demonstrate improved performance. More importantly, by analyzing the expanded space of estimators both mathematically and empirically we find that dual solutions offer greater flexibility in navigating the tradeoff between optimization stability and estimation bias, and generally provide superior estimates in practice.

## 1   Introduction

One of the most fundamental problems in reinforcement learning (RL) is *policy evaluation*, where we seek to estimate the expected long-term payoff of a given *target* policy in a decision making environment. An important variant of this problem, *off-policy evaluation* (OPE) [23], is motivated by applications where deploying a policy in a live environment entails significant cost or risk [20, 27]. To circumvent these issues, OPE attempts to estimate the value of a target policy by referring only to a dataset of experience previously gathered by other policies in the environment. Often, such logging or *behavior* policies are not known explicitly (*e.g.*, the experience may come from human actors), which necessitates the use of *behavior-agnostic* OPE methods [21].

While behavior-agnostic OPE appears to be a daunting problem, a number of estimators have recently been developed for this scenario [21, 28, 30, 31], demonstrating impressive empirical results. Such estimators, known collectively as the "DICE" family for *DIstribution Correction Estimation*, model the ratio between the propensity of the target policy to visit particular state-action pairs relative to their likelihood of appearing in the logged data. A distribution corrector of this form can then be directly used to estimate the value of the target policy.

Although there are many commonalities between the various DICE estimators, their derivations are distinct and seemingly incompatible. For example, *DualDICE* [21] is derived by a particular change-of-variables technique, whereas *GenDICE* [30] observes that the substitution strategy cannot work in the average reward setting, and proposes a distinct derivation based on distribution matching. *GradientDICE* [31] notes that GenDICE exacerbates optimization difficulties, and proposes a variant designed for limited sampling capabilities. Despite these apparent differences in these methods, the algorithms all involve a minimax optimization that has a strikingly similar form, which suggests that there is a common connection that underlies the alternative derivations.

We show that the previous DICE formulations are all in fact equivalent to regularized Lagrangians of the same linear program (LP). This LP shares an intimate relationship with the policy evaluation problem, and has a primal form we refer to as the $Q$-LP and a dual form we refer to as the $d$-LP. The primal form has been concurrently identified and studied in the context of policy optimization [22],

---

[*]Indicates equal contribution. Email: `{sherryy, ofirnachum, bodai}@google.com`.

but we focus on the $d$-LP formulation for off-policy evaluation here, which we find to have a more succinct and revealing form for this purpose. Using the $d$-LP, we identify a number of key choices in translating it into a *stable* minimax optimization problem – *i.e.* whether to include redundant constraints, whether to regularize the primal or dual variables – in addition to choices in how to translate an optimized solution into an *asymptotically unbiased* estimate of the policy value.[2] We use this characterization to show that the known members of the DICE family are a small subset of specific choices made within a much larger, unexplored set of potential OPE methods.

To understand the consequences of the various choices, we provide a comprehensive study. First, we theoretically investigate which configurations lead to bias in the primal or dual solutions, and when this affects the final estimates. Our analysis shows that the dual solutions offer greater flexibility in stabilizing the optimization while preserving asymptotic unbiasedness, versus primal solutions. We also perform an extensive empirical evaluation of the various choices across different domains and function approximators, and identify novel configurations that improve the observed outcomes.

## 2  Background

We consider an infinite-horizon Markov Decision Process (MDP) [24], specified by a tuple $\mathcal{M} = \langle S, A, R, T, \mu_0, \gamma \rangle$, which consists of a state space, action space, reward function, transition probability function, initial state distribution, and discount factor $\gamma \in [0, 1]$.[3] A policy $\pi$ interacts with the environment starting at an initial state $s_0 \sim \mu_0$, producing a distribution $\pi(\cdot|s_t)$ over $A$ from which an action $a_t$ is sampled and applied to the environment at each step $t \geq 0$. The environment produces a scalar reward $r_t = R(s_t, a_t)$,[4] and transitions to a new state $s_{t+1} \sim T(s_t, a_t)$.

### 2.1  Policy Evaluation

The *value* of a policy $\pi$ is defined as the normalized expected per-step reward it obtains:

$$\rho(\pi) := (1 - \gamma)\mathbb{E}\left[\sum_{t=0}^{\infty} \gamma^t R(s_t, a_t) \mid s_0 \sim \mu_0, \forall t, a_t \sim \pi(s_t), s_{t+1} \sim T(s_t, a_t)\right]. \quad (1)$$

In the policy evaluation setting, the policy being evaluated is referred to as the *target* policy. The value of a policy may be expressed in two equivalent ways:

$$\rho(\pi) = (1 - \gamma) \cdot \mathbb{E}_{\substack{a_0 \sim \pi(s_0) \\ s_0 \sim \mu_0}}[Q^\pi(s_0, a_0)] = \mathbb{E}_{(s,a) \sim d^\pi}[R(s, a)], \quad (2)$$

where $Q^\pi$ and $d^\pi$ are the *state-action values* and *visitations* of $\pi$, respectively, which satisfy

$$Q^\pi(s, a) = R(s, a) + \gamma \cdot \mathcal{P}^\pi Q^\pi(s, a), \text{ where } \mathcal{P}^\pi Q(s, a) := \mathbb{E}_{s' \sim T(s,a), a' \sim \pi(s')}[Q(s', a')], \quad (3)$$

$$d^\pi(s, a) = (1 - \gamma)\mu_0(s)\pi(a|s) + \gamma \cdot \mathcal{P}^\pi_* d^\pi(s, a), \text{ where } \mathcal{P}^\pi_* d(s, a) := \pi(a|s) \sum_{\tilde{s}, \tilde{a}} T(s|\tilde{s}, \tilde{a}) d(\tilde{s}, \tilde{a}). \quad (4)$$

Note that $\mathcal{P}^\pi$ and $\mathcal{P}^\pi_*$ are linear operators that are transposes (adjoints) of each other. We refer to $\mathcal{P}^\pi$ as the *policy transition operator* and $\mathcal{P}^\pi_*$ as the *transpose policy transition operator*. The function $Q^\pi$ corresponds to the $Q$-values of the policy $\pi$; it maps state-action pairs $(s, a)$ to the expected value of policy $\pi$ when run in the environment starting at $(s, a)$. The function $d^\pi$ corresponds to the on-policy distribution of $\pi$; it is the normalized distribution over state-action pairs $(s, a)$ measuring the likelihood $\pi$ enounters the pair $(s, a)$, averaging over time via $\gamma$-discounting. We make the following standard assumption, which is common in previous policy evaluation work [30, 22].

**Assumption 1** (MDP ergodicity). *There is unique fixed point solution to* (4).

When $\gamma \in [0, 1)$, (4) always has a unique solution, as 0 cannot belong to the spectrum of $I - \gamma \mathcal{P}^\pi_*$. For $\gamma = 1$, the assumption reduces to ergodicity for discrete case under a restriction of $d$ to a normalized distribution; the continuous case is treated by [18].

### 2.2  Off-policy Evaluation via the DICE Family

Off-policy evaluation (OPE) aims to estimate $\rho(\pi)$ using only a *fixed* dataset of experiences. Specifically, we assume access to a finite dataset $\mathcal{D} = \{(s_0^{(i)}, s^{(i)}, a^{(i)}, r^{(i)}, s'^{(i)})\}_{i=1}^N$, where $s_0^{(i)} \sim \mu_0$,

$(s^{(i)}, a^{(i)}) \sim d^{\mathcal{D}}$ are samples from some unknown distribution $d^{\mathcal{D}}$, $r^{(i)} = R(s^{(i)}, a^{(i)})$, and $s'^{(i)} \sim T(s^{(i)}, a^{(i)})$. We at times abuse notation and use $(s, a, r, s') \sim d^{\mathcal{D}}$ or $(s, a, r) \sim d^{\mathcal{D}}$ as a shorthand for $(s, a) \sim d^{\mathcal{D}}, r = R(s, a), s' \sim T(s, a)$, which simulates sampling from the dataset $\mathcal{D}$ when using a finite number of samples.

The recent DICE methods take advantage of the following expression for the policy value:
$$\rho(\pi) = \mathbb{E}_{(s,a,r)\sim d^{\mathcal{D}}}\left[\zeta^*(s, a) \cdot r\right], \tag{5}$$
where $\zeta^*(s, a) := d^\pi(s, a)/d^{\mathcal{D}}(s, a)$ is the *distribution correction ratio*. The existing DICE estimators seek to approximate this ratio without knowledge of $d^\pi$ or $d^{\mathcal{D}}$, and then apply (5) to derive an estimate of $\rho(\pi)$. This general paradigm is supported by the following assumption.

**Assumption 2** (Boundedness)**.** *The stationary correction ratio is bounded,* $\|\zeta^*\|_\infty \leq C < \infty$.

When $\gamma < 1$, DualDICE [21] chooses a convex objective whose optimal solution corresponds to this ratio, and employs a change of variables to transform the dependence on $d^\pi$ to $\mu_0$. GenDICE [30], on the other hand, minimizes a divergence between successive on-policy state-action distributions, and introduces a normalization constraint to ensure the estimated ratios average to 1 over the off-policy dataset. Both DualDICE and GenDICE apply Fenchel duality to reduce an intractable convex objective to a minimax objective, which enables sampling and optimization in a stochastic or continuous action space. GradientDICE [31] extends GenDICE by using a linear parametrization so that the minimax optimization is convex-concave with convergence guarantees.

# 3 A Unified Framework of DICE Estimators

In this section, given a fixed target policy $\pi$, we present a linear programming representation (LP) of its state-action stationary distribution $d^\pi(s, a) \in \mathcal{P}$, referred to as the $d$-LP. Here $\mathcal{P}$ represents the space of all stationary distributions. The dual of this LP has solution $Q^\pi$, thus revealing the duality between the $Q$-function and the $d$-distribution of any policy $\pi$. We then discuss the mechanisms by which one can improve optimization stability through the application of regularization and redundant constraints. Although in general this may introduce bias into the final value estimate, there are a number of valid configurations for which the resulting estimator for $\rho(\pi)$ remains *unbiased*. We show that existing DICE algorithms cover several choices of these configurations, while there is also a sizable subset which remains unexplored.

## 3.1 Linear Programming Representation for the $d^\pi$-distribution

The following theorem presents a formulation of $\rho(\pi)$ in terms of a linear program with respect to the constraints in (4) and (3).

**Theorem 1.** *Given a policy $\pi$, under Assumption 1, its value $\rho(\pi)$ defined in (1) can be expressed by the following $d$-LP:*
$$\max_{d:S \times A \to \mathbb{R}} \mathbb{E}_d\left[R(s, a)\right], \quad \text{s.t.,} \quad d(s, a) = \underbrace{(1-\gamma)\mu_0(s)\pi(a|s) + \gamma \cdot \mathcal{P}_*^\pi d(s, a)}_{\mathcal{B}_*^\pi \cdot d}. \tag{6}$$
*We refer to the $d$-LP above as the **dual** problem. Its corresponding **primal** LP is*
$$\min_{Q:S \times A \to \mathbb{R}} (1-\gamma) \mathbb{E}_{\mu_0 \pi}\left[Q(s, a)\right], \quad \text{s.t.,} \quad Q(s, a) = \underbrace{R(s, a) + \gamma \cdot \mathcal{P}^\pi Q(s, a)}_{\mathcal{B}^\pi \cdot Q}. \tag{7}$$

*Proof.* Notice that under Assumption 1, the constraint in (6) determines a unique solution, which is the stationary distribution $d^\pi$. Therefore, the objective will be $\rho(\pi)$ by definition. On the other hand, due to the contraction of $\gamma \cdot \mathcal{P}^\pi$, the primal problem is feasible and the solution is $Q^\pi$, which shows the optimal objective value will also be $\rho(\pi)$, implying strong duality holds. $\qquad\square$

Theorem 1 presents a succinct LP representation for policy value and reveals the duality between the $Q^\pi$-function and $d^\pi$-distribution, thus providing an answer to the question raised by [28]. Although the $d$-LP provides a mechanism for policy evaluation, directly solving either the primal or dual $d$-LPs is difficult due to the number of constraints, which will present difficulties when the state and action spaces is uncountable. These issues are exaggerated in the off-policy setting where one only has access to samples $(s_0, s, a, r, s')$ from a stochastic process. To overcome these difficulties, one can

instead approach these primal and dual LPs through the Lagrangian,

$$\max_d \min_Q L(d, Q) := (1 - \gamma) \cdot \mathbb{E}_{\substack{a_0 \sim \pi(s_0) \\ s_0 \sim \mu_0}}[Q(s_0, a_0)] + \sum_{s,a} d(s, a) \cdot (R(s, a) + \gamma \mathcal{P}^\pi Q(s, a) - Q(s, a)).$$

In order to enable the use of an arbitrary off-policy distribution $d^\mathcal{D}$, we make the change of variables $\zeta(s, a) := d(s, a)/d^\mathcal{D}(s, a)$. This yields an equivalent Lagrangian in a more convenient form:

$$
\begin{aligned}
\max_\zeta \min_Q L_D(\zeta, Q) := &(1 - \gamma) \cdot \mathbb{E}_{\substack{a_0 \sim \pi(s_0) \\ s_0 \sim \mu_0}}[Q(s_0, a_0)] \\
&+ \mathbb{E}_{\substack{(s,a,r,s') \sim d^\mathcal{D} \\ a' \sim \pi(s')}}[\zeta(s, a) \cdot (r + \gamma Q(s', a') - Q(s, a))].
\end{aligned}
\tag{8}
$$

The Lagrangian has primal and dual solutions $Q^* = Q^\pi$ and $\zeta^* = d^\pi/d^\mathcal{D}$. Approximate solutions to one or both of $\hat{Q}, \hat{\zeta}$ can be used to estimate $\hat{\rho}(\pi)$, by either using the standard DICE paradigm in (5) which corresponds to the dual objective in (6) or, alternatively, by using the primal objective in (7) or the Lagrangian objective in (8); we further discuss these choices later in this section. Although the Lagrangian in (8) should in principle be able to derive the solutions $Q^\pi, d^\pi$ and so yield accurate estimates of $\rho(\pi)$, in practice there are a number of optimization difficulties that are liable to be encountered. Specifically, even in tabular case, due to lack of curvature, the Lagrangian is not strongly-convex-strongly-concave, and so one cannot guarantee the convergence of the final solution with stochastic gradient descent-ascent (SGDA). These optimization issues can become more severe when moving to the continuous case with neural network parametrization, which is the dominant application case in practice. In order to mitigate these issues, we present a number of ways to introduce more stability into the optimization and discuss how these mechanisms may trade-off with the bias of the final estimate. We will show that the application of certain mechanisms recovers the existing members of the DICE family, while a larger set remains unexplored.

### 3.2 Regularizations and Redundant Constraints

The augmented Lagrangian method (ALM) [25] is proposed exactly for circumventing the optimization instability, where strong convexity is introduced by adding extra regularizations *without* changing the optimal solution. However, directly applying ALM, *i.e.*, adding $h_p(Q) := \|\mathcal{B}^\pi \cdot Q - Q\|_{d^\mathcal{D}}^2$ or $h_d(d) := D_f(d\|\mathcal{B}_*^\pi \cdot d)$ where $D_f$ denotes the $f$-divergence, will introduce extra difficulty, both statistically and algorithmically, due to the conditional expectation operator in $\mathcal{B}^\pi$ and $\mathcal{B}_*^\pi$ inside of the non-linear function in $h_p(Q)$ and $h_d(d)$, which is known as "double sample" in the RL literature [1]. Therefore, the vanilla stochastic gradient descent is no longer applicable [5], due to the bias in the gradient estimator.

In this section, we use the spirit of ALM but explore other choices of regularizations to introduce strong convexity to the original Lagrangian (8). In addition to regularizations, we also employ the use of redundant constraints, which serve to add more structure to the optimization without affecting the optimal solutions. We will later analyze for which configurations these modifications of the original problem will lead to biased estimates for $\rho(\pi)$.

We first present the unified objective in full form equipped with all choices of regularizations and redundant constraints:

$$
\begin{aligned}
\max_{\zeta \geq 0} \min_{Q, \lambda} L_D(\zeta, Q, \lambda) := &(1 - \gamma) \cdot \mathbb{E}_{\substack{a_0 \sim \pi(s_0) \\ s_0 \sim \mu_0}}[Q(s_0, a_0)] + \lambda \\
&+ \mathbb{E}_{\substack{(s,a,r,s') \sim d^\mathcal{D} \\ a' \sim \pi(s')}}[\zeta(s, a) \cdot (\alpha_R \cdot R(s, a) + \gamma Q(s', a') - Q(s, a) - \lambda)] \\
&+ \alpha_Q \cdot \mathbb{E}_{(s,a) \sim d^\mathcal{D}}[f_1(Q(s, a))] - \alpha_\zeta \cdot \mathbb{E}_{(s,a) \sim d^\mathcal{D}}[f_2(\zeta(s, a))].
\end{aligned}
\tag{9}
$$

Now, let us explain each term in $(\alpha_Q, \alpha_\zeta, \alpha_R, \zeta \geq 0, \lambda)$.

- **Primal** and **Dual** **regularization:** To introduce better curvature into the Lagrangian, we introduce primal and dual regularization $\alpha_Q \mathbb{E}_{d^\mathcal{D}}[f_1(Q)]$ or $\alpha_\zeta \mathbb{E}_{d^\mathcal{D}}[f_2(\zeta)]$, respectively. Here $f_1, f_2$ are some convex and lower-semicontinuous functions.

- **Reward:** Scaling the reward may be seen as an extension of the dual regularizer, as it is a component in the dual objective in (6). We consider $\alpha_R \in \{0, 1\}$.

- **Positivity:** Recall that the solution to the original Lagrangian is $\zeta^*(s, a) = \frac{d^\pi(s,a)}{d^\mathcal{D}(s,a)} \geq 0$. We thus consider adding a positivity constraint to the dual variable. This may be interpreted as modifying the original $d$-LP in (6) to add a condition $d \geq 0$ to its objective.

- **Normalization:** Similarly, the normalization constraint also comes from the property of the optimal solution $\zeta^*(s,a)$, *i.e.*, $\mathbb{E}_{d^{\mathcal{D}}}[\zeta(s,a)] = 1$. If we add an extra constraint to the $d$-LP (6) as $\sum_{s,a} d(s,a) = 1$ and apply the Lagrangian, we result in the term $\lambda - \mathbb{E}_{d^{\mathcal{D}}}[\lambda\zeta(s,a)]$ seen in (9).

As we can see, the latter two options come from the properties of optimal dual solution, and this suggests that their inclusion would not affect the optimal dual solution. On the other hand, the first two options (primal/dual regularization and reward scaling) will in general affect the solutions to the optimization. Whether a bias in the solution affects the final estimate depends on the estimator being used.

**Remark (Robust optimization justification):** Besides the motivation from ALM for strong convexity, the regularization terms in (9), $\alpha_Q \cdot \mathbb{E}_{(s,a)\sim d^{\mathcal{D}}}[f_1(Q(s,a))]$ and $\alpha_\zeta \cdot \mathbb{E}_{(s,a)\sim d^{\mathcal{D}}}[f_2(\zeta(s,a))]$, can also be interpreted as introducing robustness via perturbation to the Bellman difference. Consider dual regularization as an example. The Fenchel dual of $\alpha_\zeta \cdot \mathbb{E}_{(s,a)\sim d^{\mathcal{D}}}[f_2(\zeta(s,a))]$ gives $\alpha_\zeta \left\{\max_{\delta(s,a)\in\Omega} \langle \zeta, \delta \rangle - \mathbb{E}_{(s,a)\sim d^{\mathcal{D}}}[f_2^*(\delta(s,a))]\right\}$, where $\Omega$ denotes the domain of function $f_2^*$. By plugging $f_2^*(\cdot) = (\cdot)^2$ back into (9), we obtain

$$\max_{\zeta\geq 0} \min_{Q,\lambda,\delta\in\Omega} L_D(\zeta,Q,\lambda) := (1-\gamma)\cdot\mathbb{E}_{\substack{a_0\sim\pi(s_0)\\s_0\sim\mu_0}}[Q(s_0,a_0)] + \lambda$$
$$+ \mathbb{E}_{\substack{(s,a,r,s')\sim d^{\mathcal{D}}\\a'\sim\pi(s')}}[\zeta(s,a)\cdot(\alpha_R\cdot R(s,a) + \gamma Q(s',a') - Q(s,a) - \lambda - \alpha_\zeta\delta(s,a))]$$
$$+ \alpha_Q \cdot \mathbb{E}_{(s,a)\sim d^{\mathcal{D}}}[f_1(Q(s,a))] + \alpha_\zeta \cdot \mathbb{E}_{(s,a)\sim d^{\mathcal{D}}}[\delta^2(s,a)], \tag{10}$$

which can be understood as introducing slack variables (i.e., perturbations in the $L_2$-ball) to the Bellman difference $\alpha_R \cdot R(s,a) + \gamma Q(s',a') - Q(s,a)$. Different dual regularizations will result in perturbation in different dual spaces. From this perspective, the dual regularization mitigates both sampling error in approximating the Bellman difference and approximation error induced by the parametrization of $Q$. Similarly, the primal regularization can be interpreted as introducing slack variables to the stationary state-action distribution condition. Please refer to Appendix A for details.

Given estimates $\hat{Q}, \hat{\lambda}, \hat{\zeta}$, there are three potential ways to estimate $\rho(\pi)$.

- **Primal estimator:** $\hat{\rho}_Q(\pi) := (1-\gamma)\cdot\mathbb{E}_{\substack{a_0\sim\pi(s_0)\\s_0\sim\mu_0}}[\hat{Q}(s_0,a_0)] + \hat{\lambda}$.

- **Dual estimator:** $\hat{\rho}_\zeta(\pi) := \mathbb{E}_{(s,a,r)\sim d^{\mathcal{D}}}[\hat{\zeta}(s,a)\cdot r]$.

- **Lagrangian:** $\hat{\rho}_{Q,\zeta}(\pi) := \hat{\rho}_Q(\pi) + \hat{\rho}_\zeta(\pi) + \mathbb{E}_{\substack{(s,a,r,s')\sim d^{\mathcal{D}}\\a'\sim\pi(s')}}\left[\hat{\zeta}(s,a)(\gamma\hat{Q}(s',a') - \hat{Q}(s,a) - \hat{\lambda})\right]$.

The following theorem outlines when a choice of regularizations, redundant constraints, and final estimator will provably result in an unbiased estimate of policy value.

**Theorem 2** (Regularization profiling). *Under Assumption 1 and 2, we summarize the effects of* $(\alpha_Q, \alpha_\zeta, \alpha_R, \zeta\geq 0, \lambda)$, *which corresponds to primal and dual regularizations, w/w.o. reward, and positivity and normalization constraints. without considering function approximation.*

| Regularization (with or without $\lambda$) | | | $\hat{\rho}_Q$ | $\hat{\rho}_\zeta$ | $\hat{\rho}_{Q,\zeta}$ |
|---|---|---|---|---|---|
| $\alpha_\zeta = 0$ $\alpha_Q > 0$ | $\alpha_R = 1$ | $\zeta$ free | **Unbiased** | Biased | **Unbiased** |
| | | $\zeta \geq 0$ | | | Biased |
| | $\alpha_R = 0$ | $\zeta$ free | Biased | | |
| | | $\zeta \geq 0$ | | | |
| $\alpha_\zeta > 0$ $\alpha_Q = 0$ | $\alpha_R = 1$ | $\zeta$ free | | **Unbiased** | **Unbiased** |
| | | $\zeta \geq 0$ | | | |
| | $\alpha_R = 0$ | $\zeta$ free | | | |
| | | $\zeta \geq 0$ | | | |

Notice that the primal and dual solutions can both be unbiased under specific regularization configurations, but the dual solutions are unbiased in 6 out of 8 such configurations, whereas the primal

solution is unbiased in only 1 configuration. The primal solution additionally requires the positivity constraint to be excluded (see details in Appendix B), further restricting its optimization choices.

The Lagrangian estimator is unbiased when at least one of $\hat{Q}$, $\hat{\lambda}$ or $\hat{\zeta}$ are unbiased. This property is referred to as *doubly robust* in the literature [12] This seems to imply that the Lagrangian estimator is optimal for behavior-agnostic off-policy evaluation. However, this is not the case as we will see in the empirical analysis. Instead, the approximate dual solutions are typically more accurate than approximate primal solutions. Since neither is exact, the Lagrangian suffers from error in both, while the dual estimator $\hat{\rho}_\zeta$ will exhibit more robust performance, as it solely relies on the approximate $\hat{\zeta}$.

### 3.3 Recovering Existing OPE Estimators

This organization provides a complete picture of the DICE family of estimators. Existing DICE estimators can simply be recovered by picking one of the valid regularization configurations:

- **DualDICE** [21]: $(\alpha_Q = 0, \alpha_\zeta = 1, \alpha_R = 0)$ without $\zeta \geq 0$ and without $\lambda$. DualDICE also derives an *unconstrained primal form* where optimization is exclusively over the primal variables (see Appendix D). This form results in a *biased* estimate but avoids difficults in minimax optimization, which again is a tradeoff between optimization stability and solution unbiasedness.
- **GenDICE** [30] and **GradientDICE** [31]: $(\alpha_Q = 1, \alpha_\zeta = 0, \alpha_R = 0)$ with $\lambda$. GenDICE differs from GradientDICE in that GenDICE enables $\zeta \geq 0$ whereas GradientDICE disables it.
- **DR-MWQL** and **MWL** [28]: $(\alpha_Q = 0, \alpha_\zeta = 0, \alpha_R = 1)$ and $(\alpha_Q = 0, \alpha_\zeta = 0, \alpha_R = 0)$, both without $\zeta \geq 0$ and without $\lambda$.
- **LSTDQ** [15]: With linear parametrization for $\zeta(s, a) = v^\top \phi(s, a)$ and $Q(s, a) = w^\top \phi(s, a)$, for any *unbiased* estimator without $\zeta \geq 0$ and $\lambda$ in Theorem 2, we can recover LSTDQ. Please refer to Appendix C for details.
- **Algae** $Q$-**LP** [22]: $(\alpha_Q = 0, \alpha_\zeta = 1, \alpha_R = 1, \zeta \geq 0)$ without $\zeta \geq 0$ and without $\lambda$.
- **BestDICE:** $(\alpha_Q = 0, \alpha_\zeta = 1, \alpha_R = 0/1)$ with $\zeta \geq 0$ and with $\lambda$. More importantly, we discover a variant that achieves the best performance, which was not identified without this unified framework.

## 4 Experiments

In this section, we empirically verify the theoretical findings. We evaluate different choices of estimators, regularizers, and constraints, on a set of OPE tasks ranging from tabular (Grid) to discrete-control (Cartpole) and continuous-control (Reacher), under linear and neural network parametrizations, with offline data collected from behavior policies with different noise levels ($\pi_1$ and $\pi_2$). See Appendix F for implementation details and additional results. Our empirical conclusions are as follows:

- The dual estimator $\hat{\rho}_\zeta$ is unbiased under more configurations and yields best performance out of all estimators, and furthermore exhibits strong robustness to scaling and shifting of MDP rewards.
- Dual regularization ($\alpha_\zeta > 0$) yields better estimates than primal regularization; the choice of $\alpha_R \in \{0, 1\}$ exhibits a slight advantage to $\alpha_R = 1$.
- The inclusion of redundant constraints ($\lambda$ and $\zeta \geq 0$) improves stability and estimation performance.
- As expected, optimization using the unconstrained primal form is more stable but also more biased than optimization using the minimax regularized Lagrangian.

Based on these findings, we propose a particular set of choices that generally performs well, overlooked by previously proposed DICE estimators: the dual estimator $\hat{\rho}_\zeta$ with regularized dual variable ($\alpha_\zeta > 0, \alpha_R = 1$) and redundant constraints ($\lambda, \zeta \geq 0$) optimized with the Lagrangian.

### 4.1 Choice of Estimator ($\hat{\rho}_Q$, $\hat{\rho}_\zeta$, or $\hat{\rho}_{Q,\zeta}$)

We first consider the choice of estimator. In each case, we perform Lagrangian optimization with regularization chosen according to Theorem 2 to not bias the resulting estimator. We also use $\alpha_R = 1$ and include redundant constraints for $\lambda$ and $\zeta \geq 0$ in the dual estimator. Although not shown, we also evaluated combinations of regularizations which can bias the estimator (as well as no regularizations) and found that these generally performed worse; see Section 4.2 for a subset of these experiments.

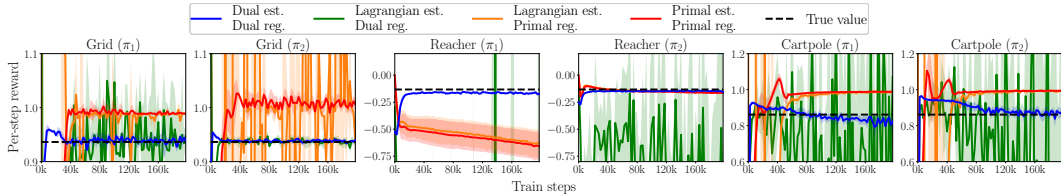

Figure 1: Estimation results on Grid, Reacher, and Cartpole using data collected from different behavior policies ($\pi_2$ is closer to the target policy than $\pi_1$). Biased estimator-regularizer combinations from Theorem 2 are omitted. The dual estimator with regularized dual variable outperforms all other estimators/regularizers. Lagrangian can be as good as the dual but has a larger variance.

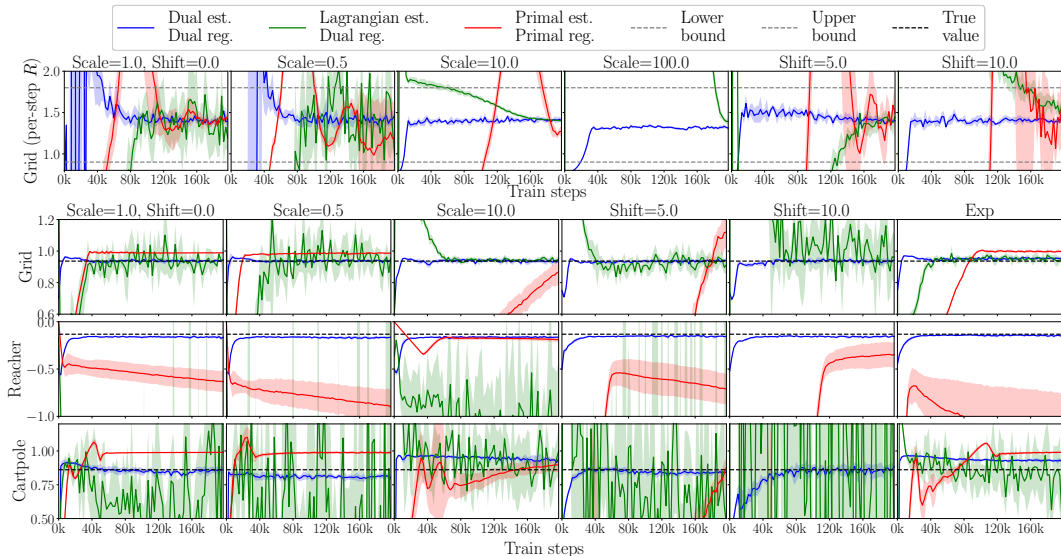

Figure 2: Primal (red), dual (blue), and Lagrangian (green) estimates under linear (top) and neural network (bottom) parametrization when rewards are transformed during training. Estimations are transformed back and plotted on the original scale. The dual estimates are robust to all transformations, whereas the primal and Lagrangian estimates are sensitive to the reward values.

Our evaluation of different estimators is presented in Figure 1. We find that the dual estimator consistently produces the best estimates across different tasks and behavior policies. In comparison, the primal estimates are significantly worse. While the Lagrangian estimator can improve on the primal, it generally exhibits higher variance than the dual estimator. Presumably, the Lagrangian does not benefit from the doubly robust property, since both solutions are biased in this practical setting.

To more extensively evaluate the dual estimator, we investigate its performance when the reward function is scaled by a constant, shifted by a constant, or exponentiated. [5] To control for difficulties in optimization, we first parametrize the primal and dual variables as linear functions, and use stochastic gradient descent to solve the convex-concave minimax objective in (9) with $\alpha_Q = 0$, $\alpha_\zeta = 1$, and $\alpha_R = 1$. Since a linear parametrization changes the ground truth of evaluation, we compute the upper and lower estimation bounds by only parameterizing the primal or the dual variable as a linear function. Figure 2 (top) shows the estimated per-step reward of the Grid task. When the original reward is used (col. 1), the primal, dual, and Lagrangian estimates eventually converge to roughly the same value (even though primal estimates converge much slower). When the reward is scaled by 10 or 100 times or shifted by 5 or 10 units (the original reward is between 0 and 1), the resulting primal estimates are severely affected and do not converge given the same number of gradient updates. When performing this same evaluation with neural network parametrization (Figure 2, bottom), the primal estimates continue to exhibit sensitivity to reward transformations, whereas the dual estimates stay roughly the same after being transformed back to the original scale. We further implemented target

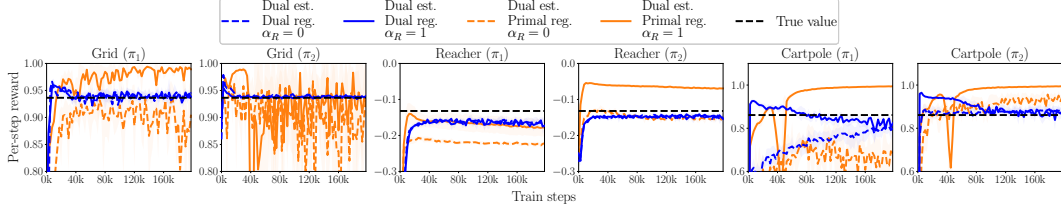

Figure 3: Dual estimates when $\alpha_R = 0$ (dotted line) and $\alpha_R = 1$ (solid line). Regularizing the dual variable (blue) is consistently better than regularizing the primal variable (orange). $\alpha_R \neq 0$ and $\alpha_Q \neq 0$ leads to biased estimation (solid orange). The value of $\alpha_R$ does not affect the final estimate when $\alpha_\zeta = 1, \alpha_Q = 0$.

network for training stability of the primal variable, and the same concolusion holds (see Appendix). Note that while the dual solution is robust to the scale and range of rewards, the optimization objective used here still has $\alpha_R = 1$, which is different from $\alpha_R = 0$ where $\hat{\rho}_Q$ is no longer a valid estimator.

## 4.2   Choice of Regularization ($\alpha_\zeta$, $\alpha_R$, and $\alpha_Q$)

Next, we study the choice between regularizing the primal or dual variables. Given the results of Section 4.1, we focus on ablations using the dual estimator $\hat{\rho}_\zeta$ to estimate $\rho_\pi$. Results are presented in Figure 3. As expected, we see that regularizing the primal variables when $\alpha_R = 1$ leads to a biased estimate, especially in Grid ($\pi_1$), Reacher ($\pi_2$), and Cartpole. Regularizing the dual variable (blue lines) on the other hand does not incur such a bias. Additionally, the value of $\alpha_R$ has little effect on the final estimates when the dual variable is regularized (dotted versus solid blue lines). While the invariance to $\alpha_R$ may not generalize to other tasks, an advantage of the dual estimates with regularized dual variable is the flexibility to set $\alpha_R = 0$ or $1$ depending on the reward function.

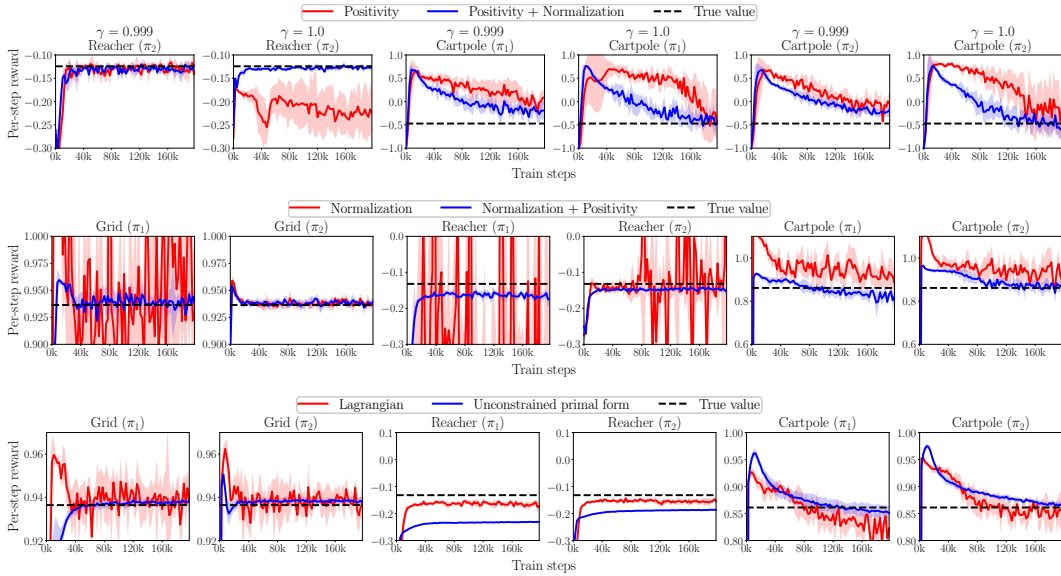

Figure 4: Apply positive constraint, normalization constraint, and the unconstrained primal form during optimization (blue curves). Positivity constraint (row 1) improves training stability. Normalization constraint is essential when $\gamma = 1$, and also helps when $\gamma < 1$ (row 2). Solving the unconstrained primal problem (row 3) can be useful when the action space is discrete.

## 4.3   Choice of Redundant Constraints ($\lambda$ and $\zeta \geq 0$)

So far our experiments with the dual estimator used $\lambda$ and $\zeta \geq 0$ in the optimizations, corresponding to the normalization and positive constraints in the $d$-LP. However, these are in principle not necessary

when $\gamma < 1$, and so we evaluate the effect of removing them. Given the results of the previous sections, we focus our ablations on the use of the dual estimator $\hat{\rho}_\zeta$ with dual regularization $\alpha_\zeta > 0, \alpha_R = 1$.

**Normalization.** We consider the effect of removing the normalization constraint ($\lambda$). Figure 4 (row 1) shows the effect of keeping (blue curve) or removing (red curve) this constraint during training. We see that training becomes less stable and approximation error increases, even when $\gamma < 1$.

**Positivity.** We continue to evaluate the effect of removing the positivity constraint $\zeta \geq 0$, which, in our previous experiments, was enforced via applying a square function to the dual variable neural network output. Results are presented in Figure 4 (row 2), where we again see that the removal of this constraint is detrimental to optimization stability and estimator accuracy.

### 4.4 Choice of Optimization (Lagrangian or Unconstrained Primal Form)

So far, our experiments have used minimax optimization via the Lagrangian to learn primal and dual variables. We now consider solving the unconstrained primal form of the $d$-LP, which Section 3.2 suggests may lead to an easier, but biased, optimization. Figure 4 (row 3) indeed shows that the unconstrained primal reduces variance on Grid and produces better estimates on Cartpole. Both environments have discrete action spaces. Reacher, on the other hand, has a continuous action space, which creates difficulty when taking the expectation over next step samples, causing bias in the unconstrained primal form. Given this mixed performance, we generally advocate for the Lagrangian, unless the task is discrete-action and the stochasticity of the dynamics is known to be low.

## 5 Related work

Off-policy evaluation has long been studied in the RL literature [9, 12, 13, 19, 23, 27]. While some approaches are model-based [10], or work by estimating the value function [8], most rely on importance reweighting to transform the off-policy data distribution to the on-policy target distribution. They often require to know or estimate the behavior policy, and suffer a variance exponential in the horizon, both of which limit their applications. Recently, a series of works were proposed to address these challenges [14, 17, 26]. Among them is the DICE family [21, 30, 31], which performs some form of stationary distribution estimation. The present paper develops a convex duality framework that unifies many of these algorithms, and offers further important insights. Many OPE algorithms may be understood to correspond to the categories considered here. Naturally, the recent stationary distribution correction algorithms [21, 30, 31], are the dual methods. The FQI-style estimator [8] loosely corresponds to our primal estimator. Moreover, Lagrangian-type estimators are also considered [26, 28], although some are not for the behavior-agnostic setting [26].

Convex duality has been widely used in machine learning, and in RL in particular. In one line of literature, it was used to solve the Bellman equation, whose fixed point is the value function [6, 7, 16]. Here, duality facilitates derivation of an objective function that can be conveniently approximated by sample averages, so that solving for the fixed point is converted to that of finding a saddle point. Another line of work, more similar to the present paper, is to optimize the Lagrangian of the linear program that characterizes the value function [2, 4, 29]. In contrast to our work, these algorithms typically do not incorporate off-policy correction, but assume the availability of on-policy samples.

## 6 Conclusion

We have proposed a unified view of off-policy evaluation via the regularized Lagrangian of the $d$-LP. Under this unification, existing DICE algorithms are recovered by specific (suboptimal) choices of regularizers, (redundant) constraints, and ways to convert optimized solutions to policy values. By systematically studying the mathematical properties and empirical effects of these choices, we have found that the dual estimates (i.e., policy value in terms of the state-action distribution) offer greater flexibility in incorporating optimization stablizers while preserving asymptotic unibasedness, in comparison to the primal estimates (i.e., estimated $Q$-values). Our study also reveals alternative estimators not previously identified in the literature that exhibit improved performance. Overall, these findings suggest promising new directions of focus for OPE research in the offline setting.

## Broader Impact

One of the broader issues in reinforcement learning is reproducibility — it is difficult to make a compelling case that any new algorithm is actually an improvement without a common framework for reliably reproducing past related research results. This paper unifies existing DICE estimators and offers a standard implementation of a number of seemingly distinct algorithms that in fact only differ in regularization. Future researchers and practioners in this domain can benefit from this unification, where algorithms can be effectively compared in isolation from other artifacts such as neural network architecture.

Meanwhile, this research has revealed a number of choices in regularization, redundant constraints, and final estimators in DICE, suggesting a large hyperparameter search space. While we try our best to illustrate which choices matter through mathematical analysis and ablation study, individuals with limited computational resources could still be put at a disadvantage when tuning their algorithms to achieve the best performance for a specific task.

### Acknowledgments

We thank Hanjun Dai and other members of the Google Brain team for helpful discussions.

## Footnotes

[2]With a slight terminology abuse, we use asymptotically unbiased and unbiased interchangeably.

[3]For simplicity, we focus on the discounted case where $\gamma \in [0, 1)$ unless otherwise specified. The same conclusions generally hold for the undiscounted case with $\gamma = 1$; see Appendix E for more details.

[4]We consider a a deterministic reward function. All of our results apply to stochastic rewards as well.

[5]Note this is separate from $\alpha_R$, which only affects optimization. We use $\alpha_R = 1$ exclusively here.

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
