[Supplementary Material]

# Appendix

## A Robustness Justification

We explain the robustness interpretation of the dual regularization as the perturbation of Bellman differences. In this section, we elaborate the robustness interpretation of the primal regularization. For simplicity, we also consider $f_1(\cdot) = (\cdot)^2$. Therefore, we have $\alpha_Q \cdot \mathbb{E}_{(s,a)\sim d^{\mathcal{D}}}[f_1(Q(s,a))] = \alpha_Q \cdot \left\{ \max_{\delta(s,a)} \langle Q, \delta \rangle - \mathbb{E}_{(s,a)\sim d^{\mathcal{D}}}\left[\delta^2(s,a)\right] \right\}$. Plug the dual form into (9) and with strong duality, we have

$$\max_{\zeta \geq 0, \delta} \min_{Q, \lambda} L_D(\zeta, Q, \lambda, \delta) := (1-\gamma) \cdot \mathbb{E}_{\substack{a_0 \sim \pi(s_0) \\ s_0 \sim \mu_0}}[Q(s_0, a_0)] + \alpha_Q \mathbb{E}_{(s,a)\sim d^{\mathcal{D}}}[\delta(s,a) \cdot Q(s,a)] + \lambda$$
$$+ \mathbb{E}_{\substack{(s,a,r,s')\sim d^{\mathcal{D}} \\ a'\sim\pi(s')}}[\zeta(s,a) \cdot (\alpha_R \cdot R(s,a) + \gamma Q(s',a') - Q(s,a) - \lambda)]$$
$$- \alpha_Q \cdot \mathbb{E}_{(s,a)\sim d^{\mathcal{D}}}[\delta^2(s,a)] - \alpha_\zeta \cdot \mathbb{E}_{(s,a)\sim d^{\mathcal{D}}}[f_2(\zeta(s,a))], \qquad (11)$$

which can be understood as the Lagrangian of

$$\max_{\zeta \geq 0, \delta} \quad \alpha_R \mathbb{E}_{(s,a)\sim d^{\mathcal{D}}}[\zeta(s,a) \cdot R(s,a)] - \alpha_Q \cdot \mathbb{E}_{(s,a)\sim d^{\mathcal{D}}}[\delta^2(s,a)] - \alpha_\zeta \cdot \mathbb{E}_{(s,a)\sim d^{\mathcal{D}}}[f_2(\zeta(s,a))]$$
$$\text{s.t.} \qquad (1-\gamma)\mu_0\pi + \alpha_Q d^{\mathcal{D}} \cdot \delta + \gamma \cdot \mathcal{P}_*^\pi \cdot (d^{\mathcal{D}} \cdot \zeta) = (d^{\mathcal{D}} \cdot \zeta) \qquad (12)$$
$$\mathbb{E}_{(s,a)\sim d^{\mathcal{D}}}[\zeta] = 1.$$

As we can see, the primal regularization actually introduces $L_2$-ball perturbations to the stationary state-action distribution condition (12). For different regularization, the perturbations will be in different dual spaces. For examples, with entropy-regularization, the perturbation lies in the simplex. The corresponding optimization of (10) is

$$\min_Q \quad (1-\gamma)\mathbb{E}_{\mu_0\pi}[Q(s,a)] + \alpha_Q \cdot \mathbb{E}_{(s,a)\sim d^{\mathcal{D}}}[f_1(Q)] + \alpha_\zeta \cdot \mathbb{E}_{(s,a)\sim d^{\mathcal{D}}}\left[\delta^2(s,a)\right] \qquad (13)$$
$$\text{s.t.} \qquad Q(s,a) \geq R(s,a) + \gamma \cdot \mathcal{P}^\pi Q(s,a) - \alpha_\zeta \delta(s,a). \qquad (14)$$

In both (13) and (12), the relaxation of dual $\zeta$ in (12) does not affect the optimality of dual solution: the stationary state-action distribution is still the only solution to (12); while in (13), the relaxation of primal $Q$ will lead to different optimal primal solution. From this view, one can also justify the advantages of the dual OPE estimation.

## B Proof for Theorem 2

The full enumeration of $\alpha_Q, \alpha_\zeta, \alpha_R, \lambda$, and $\zeta \geq 0$ results in $2^5 = 32$ configurations. We note that it is enough to characterize the solutions $Q^*, \zeta^*$ under these different configurations. Clearly, the primal estimator $\hat{\rho}_Q$ is unbiased when $Q^* = Q^\pi$, and the dual estimator $\hat{\rho}_\zeta$ is unbiased when $\zeta^* = d^\pi/d^{\mathcal{D}}$. For the Lagrangian estimator $\hat{\rho}_{Q,\zeta}$, we may write it in two ways:

$$\hat{\rho}_{Q,\zeta}(\pi) = \hat{\rho}_Q(\pi) + \sum_{s,a} d^{\mathcal{D}}(s,a)\zeta(s,a)(R(s,a) + \gamma\mathcal{P}^\pi Q(s,a) - Q(s,a)) \qquad (15)$$

$$= \hat{\rho}_\zeta(\pi) + \sum_{s,a} Q(s,a)((1-\gamma)\mu_0(s)\pi(a|s) + \gamma\mathcal{P}_*^\pi d^{\mathcal{D}} \times \zeta(s,a) - d^{\mathcal{D}} \times \zeta(s,a)). \qquad (16)$$

It is clear that when $Q^* = Q^\pi$, the second term of (15) is 0 and $\hat{\rho}_{Q,\zeta}(\pi) = \rho(\pi)$. When $\zeta^* = d^\pi/d^{\mathcal{D}}$, the second term of (16) is 0 and $\hat{\rho}_{Q,\zeta}(\pi) = \rho(\pi)$. Therefore, the Lagrangian estimator is unbiased when either $Q^* = Q^\pi$ or $\zeta^* = d^\pi/d^{\mathcal{D}}$.

Now we continue to characterizing $Q^*, \zeta^*$ under different configurations. First, when $\alpha_Q = 0, \alpha_\zeta = 0$, it is clear that the solutions are always unbiased by virtue of Theorem 1 (see also [22]). When $\alpha_Q > 0, \alpha_\zeta > 0$, the solutions are in general biased. We summarize the remaining configurations (in the discounted case) of $\alpha_Q > 0, \alpha_\zeta = 0$ and $\alpha_Q = 0, \alpha_\zeta > 0$ in the table below. We provide proofs for the configurations of the shaded cells. Proofs for the rest configurations can be found in [21, 22].

*Proof.* Under our Assumptions 1 and 2, the strong duality holds for (9). We provide the proofs by checking the configurations case-by-case.

Table 1: Optimal solutions for all configurations. Configurations with new proofs are shaded in gray.

| Regularizer (w./w.o. $\lambda$) | | | Case | $Q^*(s,a)$ | $\zeta^*(s,a)$ | $L(Q^*,\zeta^*)$ |
|---|---|---|---|---|---|---|
| $\alpha_\zeta = 0$ $\alpha_Q > 0$ | $\alpha_R = 1$ | $\zeta$ free | i | $Q^\pi$ | $\frac{d^\pi}{d^\mathcal{D}} + \alpha_Q \frac{(\mathcal{I}-\gamma\mathcal{P}_*^\pi)^{-1}\left(d^\mathcal{D}\cdot f_1'(Q^\pi)\right)}{d^\mathcal{D}}$ | $\alpha_R(1-\gamma)\cdot\mathbb{E}_{\mu_0}[Q^\pi]$ $+\alpha_Q\mathbb{E}_{(s,a)\sim d^\mathcal{D}}\left[f_1\left(Q^\pi\right)\right]$ |
| | | $\zeta \geq 0$ | ii | $f_1^{*\prime}\left(\frac{1}{\alpha_Q}\left(\left(\alpha_Q f_1'\left(Q^\pi\right)+\frac{(1-\gamma)\mu_0\pi}{d^\mathcal{D}}\right)_+ - \frac{(1-\gamma)\mu_0\pi}{d^\mathcal{D}}\right)\right)$ | $\frac{1}{d^\mathcal{D}}\left(\mathcal{I}-\gamma\cdot\mathcal{P}^\pi\right)^{-1}\cdot$ $d^\mathcal{D}\left(\alpha_Q f_1'\left(Q^\pi\right)+\frac{(1-\gamma)\mu_0\pi}{d^\mathcal{D}}\right)_+$ | $(1-\gamma)\cdot\mathbb{E}_{\mu_0}[Q^*]$ $+\mathbb{E}_{d^\mathcal{D}}\left[\zeta^*(s,a)\cdot(\alpha_R\cdot r\right.$ $\left.+\gamma Q^*(s',a')-Q^*(s,a))\right]$ $+\alpha_Q\cdot\mathbb{E}_{d^\mathcal{D}}\left[f_1(Q^*(s,a))\right]$ |
| | $\alpha_R = 0$ | $\zeta$ free | iii | $f_1^{*\prime}(0)$ | | $-\alpha_Q f_1^*(0)$ |
| | | $\zeta \geq 0$ | iv | | | |
| $\alpha_\zeta > 0$ $\alpha_Q = 0$ | $\alpha_R = 1$ | $\zeta$ free | v | $-\alpha_\zeta\left(\mathcal{I}-\mathcal{P}^\pi\right)^{-1}f_2'\left(\frac{d^\pi}{d^\mathcal{D}}\right)$ $+\alpha_R Q^\pi$ [21, 22] | $\frac{d^\pi}{d^\mathcal{D}}$ [21, 22] | $\alpha_R\cdot\mathbb{E}_{(s,a,r,s')\sim d^\mathcal{D}}[r]$ $-\alpha_\zeta\cdot D_f(d^\pi\|d^\mathcal{D})$ [21, 22] |
| | | $\zeta \geq 0$ | vi | | | |
| | $\alpha_R = 0$ | $\zeta$ free | vii | | | |
| | | $\zeta \geq 0$ | viii | | | |

- **iii)-iv)** In this configuration, the regularized Lagrangian (9) becomes

$$\max_{\zeta\geq 0}\min_{Q,\lambda} L_D(\zeta,Q,\lambda) := \quad (1-\gamma)\cdot\mathbb{E}_{\substack{a_0\sim\pi(s_0)\\ s_0\sim\mu_0}}[Q(s_0,a_0)] + \alpha_Q\cdot\mathbb{E}_{(s,a)\sim d^\mathcal{D}}[f_1(Q(s,a))] + \lambda$$
$$+\mathbb{E}_{\substack{(s,a,r,s')\sim d^\mathcal{D}\\ a'\sim\pi(s')}}[\zeta(s,a)\cdot(\gamma Q(s',a')-Q(s,a)-\lambda)],$$

which is equivalent to

$$\max_{\zeta\geq 0}\min_Q L_D(\zeta,Q) = \quad \left\langle (1-\gamma)\mu_0\pi + \gamma\cdot\mathcal{P}_*^\pi\cdot\left(d^\mathcal{D}\cdot\zeta\right) - d^\mathcal{D}\cdot\zeta, Q\right\rangle + \alpha_Q\mathbb{E}_{d^\mathcal{D}}\left[f_1(Q)\right]$$
$$\text{s.t.}\quad \mathbb{E}_{d^\mathcal{D}}[\zeta] = 1. \tag{17}$$

Apply the Fenchel duality w.r.t. $Q$, we have

$$\max_\zeta \quad L_D(\zeta,Q^*) = -\alpha_Q\mathbb{E}_{d^\mathcal{D}}\left[f_1^*\left(\frac{(1-\gamma)\mu_0\pi+\gamma\cdot\mathcal{P}_*^\pi\cdot\left(d^\mathcal{D}\cdot\zeta\right)-d^\mathcal{D}\cdot\zeta}{\alpha_Q d^\mathcal{D}}\right)\right] \tag{18}$$
$$\text{s.t.}\quad \mathbb{E}_{d^\mathcal{D}}[\zeta] = 1. \tag{19}$$

If $f_1^*(\cdot)$ achieves the minimum at zero, it is obvious that

$$d^\mathcal{D}\cdot\zeta^* = (1-\gamma)\mu_0\pi + \gamma\cdot\mathcal{P}_*^\pi\cdot\left(d^\mathcal{D}\cdot\zeta^*\right) \Rightarrow d^\mathcal{D}\cdot\zeta^* = d^\pi.$$

Therefore, we have

$$L(\zeta^*,Q^*) = -\alpha_Q f_1^*(0),$$

and

$$Q^* = \operatorname*{argmax}_Q \left\langle (1-\gamma)\mu_0\pi + \gamma\cdot\mathcal{P}_*^\pi\cdot\left(d^\mathcal{D}\cdot\zeta^*\right) - d^\mathcal{D}\cdot\zeta^*, Q\right\rangle + \alpha_Q\mathbb{E}_{d^\mathcal{D}}\left[f_1(Q)\right]$$
$$= f_1^{*\prime}(0)$$

- **i)-ii)** Following the derivation in case **iii)-iv)**, we have the regularized Lagrangian as almost the same as (17) but has an extra term $\alpha_R\mathbb{E}_{d^\mathcal{D}}[\zeta\cdot R]$, *i.e.*

$$\max_\zeta\min_Q L_D(\zeta,Q) := \quad (1-\gamma)\cdot\mathbb{E}_{\substack{a_0\sim\pi(s_0)\\ s_0\sim\mu_0}}[Q(s_0,a_0)] + \alpha_Q\cdot\mathbb{E}_{(s,a)\sim d^\mathcal{D}}[f_1(Q(s,a))]$$
$$+\mathbb{E}_{\substack{(s,a,r,s')\sim d^\mathcal{D}\\ a'\sim\pi(s')}}[\zeta(s,a)\cdot(\alpha_R\cdot R(s,a)+\gamma Q(s',a')-Q(s,a))].$$

We first consider the case where the $\zeta$ is free and the normalization constraint is not enforced.

After applying the Fenchel duality w.r.t. $Q$, we have

$$\max_\zeta \quad L_D(\zeta,Q^*) = \alpha_R\left\langle d^\mathcal{D}\cdot\zeta, R\right\rangle - \alpha_Q\mathbb{E}_{d^\mathcal{D}}\left[f_1^*\left(\frac{d^\mathcal{D}\cdot\zeta-(1-\gamma)\mu_0\pi-\gamma\cdot\mathcal{P}_*^\pi\cdot\left(d^\mathcal{D}\cdot\zeta\right)}{\alpha_Q d^\mathcal{D}}\right)\right]. \tag{20}$$

We denote

$$\nu = \frac{d^\mathcal{D}\cdot\zeta - (1-\gamma)\mu_0\pi - \gamma\cdot\mathcal{P}_*^\pi\cdot\left(d^\mathcal{D}\cdot\zeta\right)}{d^\mathcal{D}}$$
$$\Rightarrow d^\mathcal{D}\cdot\zeta = (\mathcal{I}-\gamma\cdot\mathcal{P}_*^\pi)^{-1}\left((1-\gamma)\mu_0\pi + d^\mathcal{D}\cdot\nu\right),$$

and thus,

$$L_D\left(\zeta^*, Q^*\right) = \max_\nu \left\langle \left(\mathcal{I} - \gamma \cdot \mathcal{P}_*^\pi\right)^{-1} \left((1-\gamma)\,\mu_0\pi + d^{\mathcal{D}} \cdot \nu\right), \alpha_R R \right\rangle - \alpha_Q \mathbb{E}_{d^{\mathcal{D}}}\left[f_1^*\left(\frac{\nu}{\alpha_Q}\right)\right]$$

$$= \alpha_R\left(1-\gamma\right) \mathbb{E}_{\substack{a_0 \sim \pi(s_0) \\ s_0 \sim \mu_0}}\left[Q^\pi\left(s_0, a_0\right)\right] + \max_\nu \mathbb{E}_{d^{\mathcal{D}}}\left[\nu \cdot \left(Q^\pi\right)\right] - \alpha_Q \mathbb{E}_{d^{\mathcal{D}}}\left[f_1^*\left(\frac{\nu}{\alpha_Q}\right)\right],$$

$$= \alpha_R\left(1-\gamma\right) \mathbb{E}_{\substack{a_0 \sim \pi(s_0) \\ s_0 \sim \mu_0}}\left[Q^\pi\left(s_0, a_0\right)\right] + \alpha_Q \mathbb{E}_{d^{\mathcal{D}}}\left[f_1\left(Q^\pi\right)\right]$$

where the second equation comes from the fact $Q^\pi = \left(\mathcal{I} - \gamma \cdot \mathcal{P}^\pi\right)^{-1} R$ and last equation comes from Fenchel duality with $\nu^* = \alpha_Q f_1'\left(Q^\pi\right)$.

Then, we can characterize

$$\zeta^* = \frac{\left(\mathcal{I} - \gamma \cdot \mathcal{P}_*^\pi\right)^{-1}\left((1-\gamma)\,\mu_0\pi\right)}{d^{\mathcal{D}}} + \alpha_Q \frac{\left(\mathcal{I} - \gamma \cdot \mathcal{P}_*^\pi\right)^{-1}\left(d^{\mathcal{D}} \cdot f_1'\left(Q^\pi\right)\right)}{d^{\mathcal{D}}}$$

$$= \frac{d^\pi}{d^{\mathcal{D}}} + \alpha_Q \frac{\left(\mathcal{I} - \gamma \cdot \mathcal{P}_*^\pi\right)^{-1}\left(d^{\mathcal{D}} \cdot f_1'\left(Q^\pi\right)\right)}{d^{\mathcal{D}}},$$

and

$$Q^* = \left(f_1'\right)^{-1}\left(\frac{d^{\mathcal{D}} \cdot \zeta^* - (1-\gamma)\,\mu_0\pi - \gamma \cdot \mathcal{P}_*^\pi \cdot \left(d^{\mathcal{D}} \cdot \zeta^*\right)}{\alpha_Q d^{\mathcal{D}}}\right) = Q^\pi.$$

If we have the positive constraint, *i.e.*, $\zeta \geq 0$, we denote

$$\exp\left(\nu\right) = \frac{\left(\mathcal{I} - \gamma \cdot \mathcal{P}_*^\pi\right)\left(d^{\mathcal{D}} \cdot \zeta\right)}{d^{\mathcal{D}}} \Rightarrow d^{\mathcal{D}} \cdot \zeta = \left(\mathcal{I} - \gamma \cdot \mathcal{P}_*^\pi\right)^{-1} d^{\mathcal{D}} \cdot \exp\left(\nu\right),$$

then,

$$L_D\left(\zeta^*, Q^*\right) = \max_\nu \mathbb{E}_{d^{\mathcal{D}}}\left[\exp\left(\nu\right) \cdot Q^\pi\right] - \alpha_Q \mathbb{E}_{d^{\mathcal{D}}}\left[f_1^*\left(\frac{1}{\alpha_Q}\left(\exp\left(\nu\right) - \frac{(1-\gamma)\,\mu_0\pi}{d^{\mathcal{D}}}\right)\right)\right].$$

By first-order optimality condition, we have

$$\exp\left(\nu^*\right)\left(Q^\pi - f_1^{*\prime}\left(\frac{1}{\alpha_Q}\left(\exp\left(\nu\right) - \frac{(1-\gamma)\,\mu_0\pi}{d^{\mathcal{D}}}\right)\right)\right) = 0$$

$$= \exp\left(\nu^*\right) = \left(\alpha_Q f_1'\left(Q^\pi\right) + \frac{(1-\gamma)\,\mu_0\pi}{d^{\mathcal{D}}}\right)_+$$

$$\Rightarrow d^{\mathcal{D}} \cdot \zeta^* = \left(\mathcal{I} - \gamma \cdot \mathcal{P}^\pi\right)^{-1} \cdot d^{\mathcal{D}}\left(\alpha_Q f_1'\left(Q^\pi\right) + \frac{(1-\gamma)\,\mu_0\pi}{d^{\mathcal{D}}}\right)_+$$

$$\Rightarrow \zeta^* = \frac{1}{d^{\mathcal{D}}}\left(\mathcal{I} - \gamma \cdot \mathcal{P}^\pi\right)^{-1} \cdot d^{\mathcal{D}}\left(\alpha_Q f_1'\left(Q^\pi\right) + \frac{(1-\gamma)\,\mu_0\pi}{d^{\mathcal{D}}}\right)_+. \qquad (21)$$

For $Q^*$, we obtain from the Fenchel duality relationship,

$$Q^* = f_1^{*\prime}\left(\frac{1}{\alpha_Q}\left(\exp\left(\nu^*\right) - \frac{(1-\gamma)\,\mu_0\pi}{d^{\mathcal{D}}}\right)\right)$$

$$= f_1^{*\prime}\left(\frac{1}{\alpha_Q}\left(\left(\alpha_Q f_1'\left(Q^\pi\right) + \frac{(1-\gamma)\,\mu_0\pi}{d^{\mathcal{D}}}\right)_+ - \frac{(1-\gamma)\,\mu_0\pi}{d^{\mathcal{D}}}\right)\right). \qquad (22)$$

Then, the $L_D\left(\zeta^*, Q^*\right)$ can be obtained by plugging $(\zeta^*, Q^*)$ in (21) and (22). Obviously, in this case, the estimators are all biased.

As we can see, in both **i)** and **ii)**, none of the optimal dual solution $\zeta^*$ satisfies the normalization condition. Therefore, with the extra normalization constraint, the optimization will be obviously biased.

- **v)-viii)** These cases are also proved in [22] and we provide a more succinct proof here. In these configurations, whether $\alpha_R$ is involved or not does not affect the proof. We will keep this component for generality. We ignore the $\zeta \geq 0$ and $\lambda$ for simplicity, the conclusion does not affected, since the optimal solution $\zeta^*$ automatically satisfies these constraints.

Consider the regularized Lagrangian (9) with such configuration, we have

$$\min_{Q} \max_{\zeta} \; L_D(\zeta, Q) := \quad (1-\gamma) \cdot \mathbb{E}_{\substack{a_0 \sim \pi(s_0) \\ s_0 \sim \mu_0}}[Q(s_0, a_0)] - \alpha_\zeta \cdot \mathbb{E}_{(s,a) \sim d^{\mathcal{D}}}[f_2(\zeta(s,a))]$$

$$+ \mathbb{E}_{\substack{(s,a,r,s') \sim d^{\mathcal{D}} \\ a' \sim \pi(s')}}[\zeta(s,a) \cdot (\alpha_R \cdot R(s,a) + \gamma Q(s', a') - Q(s,a))]. \tag{23}$$

Apply the Fenchal duality to $\zeta$, we obtain

$$\min_{Q} \; L_D\left(\zeta^*, Q\right) := (1-\gamma) \cdot \mathbb{E}_{\substack{a_0 \sim \pi(s_0) \\ s_0 \sim \mu_0}}[Q(s_0, a_0)] + \alpha_\zeta \mathbb{E}_{d^{\mathcal{D}}}\left[ f_2^*\left( \frac{1}{\alpha_\zeta}\left(\mathcal{B}^\pi \cdot Q(s,a) - Q(s,a)\right)\right)\right], \tag{24}$$

with $\mathcal{B}^\pi \cdot Q(s,a) := \alpha_R \cdot R(s,a) + \gamma \mathcal{P}^\pi Q(s,a)$. We denote $\nu(s,a) = \mathcal{B} \cdot Q(s,a) - Q(s,a)$, then, we have

$$Q(s,a) = (\mathcal{I} - \gamma \cdot \mathcal{P}^\pi)^{-1}(\alpha_R \cdot R - \nu).$$

Plug this into (24), we have

$$
\begin{aligned}
L_D\left(\zeta^*, Q^*\right) &= \min_{\nu}\; (1-\gamma) \cdot \mathbb{E}_{\substack{a_0 \sim \pi(s_0) \\ s_0 \sim \mu_0}}\left[\left((\mathcal{I} - \gamma \cdot \mathcal{P}^\pi)^{-1}(\alpha_R \cdot R - \nu)\right)(s_0, a_0)\right] \\
&\quad + \alpha_\zeta \mathbb{E}_{d^{\mathcal{D}}}\left[ f_2^*\left(\frac{1}{\alpha_\zeta}\nu(s,a)\right)\right], \\
&= \alpha_R \mathbb{E}_{d^\pi}[R(s,a)] - \alpha_\zeta \max_{\nu}\left(\mathbb{E}_{d^\pi}\left[\frac{\nu(s_0, a_0)}{\alpha_\zeta}\right] + \mathbb{E}_{d^{\mathcal{D}}}\left[ f_2^*\left(\frac{1}{\alpha_\zeta}\nu(s,a)\right)\right]\right), \\
&= \alpha_R \mathbb{E}_{d^\pi}[R(s,a)] - \alpha_\zeta D_f\left(d^\pi \| d^{\mathcal{D}}\right) \tag{25}
\end{aligned}
$$

The second equation comes from the fact $d^\pi = (\mathcal{I} - \gamma \cdot \mathcal{P}_*^\pi)^{-1}(\mu_0 \pi)$. The last equation is by the definition of the Fenchel duality of $f$-divergence. Meanwhile, the optimal $\frac{1}{\alpha_\zeta}\nu^* = f_2'\left(\frac{d^\pi}{d^{\mathcal{D}}}\right)$. Then, we have

$$
\begin{aligned}
Q^* &= -(\mathcal{I} - \gamma \cdot \mathcal{P}^\pi)^{-1}\nu^* + (\mathcal{I} - \gamma \cdot \mathcal{P}^\pi)^{-1}(\alpha_R \cdot R) \\
&= -\alpha_\zeta(\mathcal{I} - \gamma \cdot \mathcal{P}^\pi)^{-1}f_2'\left(\frac{d^\pi}{d^{\mathcal{D}}}\right) + \alpha_R Q^\pi,
\end{aligned}
$$

and

$$
\begin{aligned}
\zeta^*(s,a) &= \operatorname*{argmax}_{\zeta} \zeta \cdot \nu^*(s,a) - \alpha_\zeta f_2(\zeta(s,a)) \\
&= f_2^{*\prime}\left(\frac{1}{\alpha_\zeta}\nu^*(s,a)\right) = \frac{d^\pi(s,a)}{d^{\mathcal{D}}(s,a)}.
\end{aligned}
$$

$\square$

## C  Recovering Existing OPE estimators

We verify the LSTDQ as a special case of the unified framework if the primal and dual are linearly parametrized, *i.e.*, $Q(s,a) = w^\top \phi(s,a)$ and $\zeta(s,a) = v^\top \phi(s,a)$, from any unbiased estimator without $\zeta \geq 0$ and $\lambda$. For simplicity, we assume the solution exists.

- When $(\alpha_Q = 1, \alpha_\zeta = 0, \alpha_R = 1)$, we have the estimator as

$$\max_{v} \min_{w} L_D(v, w) := (1-\gamma) \cdot w^\top \mathbb{E}_{\substack{a_0 \sim \pi(s_0) \\ s_0 \sim \mu_0}}[\phi(s_0, a_0)] + \alpha_Q \cdot \mathbb{E}_{(s,a) \sim d^{\mathcal{D}}}[f_1(w^\top \phi(s,a))]$$

$$+ v^\top \mathbb{E}_{\substack{(s,a,r,s') \sim d^{\mathcal{D}} \\ a' \sim \pi(s')}}[\phi(s,a) \cdot (\alpha_R \cdot R(s,a) + \gamma w^\top \phi(s', a') - w^\top \phi(s,a))].$$

Then, we have the first-order optimality condition for $v$ as

$$\mathbb{E}_{\substack{(s,a,r,s') \sim d^{\mathcal{D}} \\ a' \sim \pi(s')}}[\phi(s,a) \cdot (\alpha_R \cdot R(s,a) + \gamma w^\top \phi(s', a') - w^\top \phi(s,a))] = 0,$$

$$\Rightarrow \quad w = \underbrace{\mathbb{E}_{\substack{(s,a,r,s') \sim d^{\mathcal{D}} \\ a' \sim \pi(s')}}[\phi(s,a) \cdot (\phi(s,a) - \gamma \phi(s', a'))]^{-1}}_{\Xi} \mathbb{E}_{(s,a) \sim d^{\mathcal{D}}}[\alpha_R \cdot R(s,a) \phi(s,a)],$$

$$\Rightarrow \quad Q^*(s,a) = w^\top \phi(s,a),$$

which leads to

$$
\begin{aligned}
\hat{\rho}_Q(\pi) &= (1-\gamma) \cdot \mathbb{E}_{\substack{a_0 \sim \pi(s_0) \\ s_0 \sim \mu_0}}[\hat{Q}(s_0, a_0)] \\
&= (1-\gamma) \mathbb{E}_{\substack{a_0 \sim \pi(s_0) \\ s_0 \sim \mu_0}}[\phi(s,a)]^\top \Xi^{-1} \mathbb{E}_{(s,a) \sim d^{\mathcal{D}}}[R(s,a)\,\phi(s,a)].
\end{aligned}
$$

- When $(\alpha_Q = 0, \alpha_\zeta = 1, \alpha_R = \{0/1\})$, we have the estimator as

$$
\begin{aligned}
\max_v \min_w L_D(v,w) :=& (1-\gamma) \cdot w^\top \mathbb{E}_{\substack{a_0 \sim \pi(s_0) \\ s_0 \sim \mu_0}}[\phi(s_0, a_0)] - \alpha_\zeta \cdot \mathbb{E}_{(s,a) \sim d^{\mathcal{D}}}[f_2(v^\top \phi(s,a))] \\
&+ v^\top \mathbb{E}_{\substack{(s,a,r,s') \sim d^{\mathcal{D}} \\ a' \sim \pi(s')}}[\phi(s,a) \cdot (\alpha_R \cdot R(s,a) + \gamma w^\top \phi(s',a') - w^\top \phi(s,a))].
\end{aligned}
$$

Then, we have the first-order optimality condition as

$$
v^\top \mathbb{E}_{\substack{(s,a,r,s') \sim d^{\mathcal{D}} \\ a' \sim \pi(s')}}[\phi(s,a) \cdot (\gamma \phi(s',a') - \phi(s,a))] + (1-\gamma) \cdot \mathbb{E}_{\substack{a_0 \sim \pi(s_0) \\ s_0 \sim \mu_0}}[\phi(s_0, a_0)] = 0,
$$

which leads to

$$
v = (1-\gamma) \cdot \Xi^{-1} \mathbb{E}_{\substack{a_0 \sim \pi(s_0) \\ s_0 \sim \mu_0}}[\phi(s_0, a_0)].
$$

Therefore, the dual estimator is

$$
\begin{aligned}
\hat{\rho}_\zeta(\pi) &= \mathbb{E}_{(s,a,r) \sim d^{\mathcal{D}}}[R \cdot \phi(s,a)]^\top v \\
&= (1-\gamma) \mathbb{E}_{\substack{a_0 \sim \pi(s_0) \\ s_0 \sim \mu_0}}[\phi(s,a)]^\top \Xi^{-1} \mathbb{E}_{(s,a) \sim d^{\mathcal{D}}}[R(s,a)\,\phi(s,a)].
\end{aligned}
$$

- When $(\alpha_Q = 1, \alpha_\zeta = 0, \alpha_R = 0)$, by the conclusion for (17), we have

$$
v^\top \mathbb{E}_{\substack{(s,a,r,s') \sim d^{\mathcal{D}} \\ a' \sim \pi(s')}}[\phi(s,a) \cdot (\gamma \phi(s',a') - \phi(s,a))] + (1-\gamma) \cdot \mathbb{E}_{\substack{a_0 \sim \pi(s_0) \\ s_0 \sim \mu_0}}[\phi(s_0, a_0)] = 0,
$$

which leads to similar result as above case.

# D   Alternative Biased Form

**Unconstrained Primal Forms**   When $\alpha_\zeta > 0$ and $\alpha_Q = 0$, the form of the Lagranian can be simplified to yield an optimization over only $Q$. Then, we may simplify,

$$
\max_{\zeta(s,a)} \zeta(s,a) \cdot (\alpha_R \cdot R(s,a) + \gamma \mathcal{P}^\pi Q(s,a) - Q(s,a)) - \alpha_\zeta \cdot f_2(\zeta(s,a))
$$

$$
= \alpha_\zeta \cdot f_2^* \left( \frac{1}{\alpha_\zeta}(\alpha_R \cdot R(s,a) + \gamma \mathcal{P}^\pi Q(s,a) - Q(s,a)) \right). \quad (26)
$$

So, the Lagrangian may be equivalently expressed as an optimization over only $Q$:

$$
\min_Q (1-\gamma) \cdot \mathbb{E}_{\substack{a_0 \sim \pi(s_0) \\ s_0 \sim \mu_0}}[Q(s_0, a_0)] + \alpha_Q \cdot \mathbb{E}_{(s,a) \sim d^{\mathcal{D}}}[f_1(Q(s,a))]
$$

$$
+ \alpha_\zeta \cdot \mathbb{E}_{(s,a) \sim d^{\mathcal{D}}} \left[ f_2^* \left( \frac{1}{\alpha_\zeta}(\alpha_R \cdot R(s,a) + \gamma \mathcal{P}^\pi Q(s,a) - Q(s,a)) \right) \right]. \quad (27)
$$

We call this the *unconstrained primal form*, since optimization is now exclusively over primal variables. Still, given a solution $Q^*$, the optimal $\zeta^*$ to the original Lagrangian may be derived as,

$$
\zeta^*(s,a) = f_2^{*\prime}((\alpha_R \cdot R(s,a) + \gamma \mathcal{P}^\pi Q^*(s,a) - Q^*(s,a))/\alpha_\zeta). \quad (28)
$$

Although the unconstrained primal form is simpler, in practice it presents a disadvantage, due to inaccessibility of the transition operator $\mathcal{P}^\pi$. That is, in practice, one must resort to optimizing the primal form as

$$
\min_Q (1-\gamma) \cdot \mathbb{E}_{\substack{a_0 \sim \pi(s_0) \\ s_0 \sim \mu_0}}[Q(s_0, a_0)] + \alpha_Q \cdot \mathbb{E}_{(s,a) \sim d^{\mathcal{D}}}[f_1(Q(s,a))]
$$

$$
+ \alpha_\zeta \cdot \mathbb{E}_{\substack{(s,a,r,s') \sim d^{\mathcal{D}} \\ a' \sim \pi(s')}} \left[ f_2^* \left( \frac{1}{\alpha_\zeta}(\alpha_R \cdot R(s,a) + \gamma Q(s',a') - Q(s,a)) \right) \right]. \quad (29)
$$

This is in general a *biased* estimate of the true objective and thus leads to biased solutions, as the expectation over the next step samples are taken inside a square function (we choose $f_2$ to be the square function). Still, in some cases (e.g., in simple and discrete environments), the bias may be desirable as a trade-off in return for a simpler optimization.

**Unconstrained Dual Form** We have presented an unconstrained primal form. Similarly, we can derive the unconstrained dual form by removing the primal variable with a particular primal regularization $\alpha_Q \mathbb{E}_{d^{\mathcal{D}}}[f_1(Q)]$. Then, we can simplify

$$\min_{Q(s',a')} \frac{1}{d^{\mathcal{D}}(s',a')}(1-\gamma)\mu_0(s')\pi(a'|s') \cdot Q(s',a') + \alpha_Q f_1(Q)$$

$$+ \frac{1}{d^{\mathcal{D}}(s',a')}\left(\gamma \int P^{\pi}(s',a'|s,a)\, d^{\mathcal{D}} \cdot \zeta(s,a)\, ds da - d^{\mathcal{D}}(s',a')\zeta(s',a')\right) \cdot Q(s',a')$$

$$= -\alpha_Q \cdot f_1^*\left(\frac{d^{\mathcal{D}} \cdot \zeta - (1-\gamma)\mu_0\pi - \gamma\left(\mathcal{P}_*^{\pi} \cdot d^{\mathcal{D}}\right)\zeta}{\alpha_Q d^{\mathcal{D}}}\right), \tag{30}$$

with $Q^* = f_1^{*\prime}\left(\frac{d^{\mathcal{D}} \cdot \zeta - (1-\gamma)\mu_0\pi - \gamma\left(\mathcal{P}_*^{\pi} \cdot d^{\mathcal{D}}\right)\zeta}{\alpha_Q d^{\mathcal{D}}}\right)$.

So, the regularized Lagrangian can be represented as

$$\max_d \alpha_R \mathbb{E}_{d^{\mathcal{D}}}[\zeta \cdot R]$$

$$- \alpha_Q \mathbb{E}_{d^{\mathcal{D}}}\left[f_1^*\left(\frac{d^{\mathcal{D}} \cdot \zeta - (1-\gamma)\mu_0\pi - \gamma\left(\mathcal{P}_*^{\pi} \cdot d^{\mathcal{D}}\right)\zeta}{\alpha_Q d^{\mathcal{D}}}\right)\right] - \alpha_\zeta \mathbb{E}_{d^{\mathcal{D}}}[f_2(\zeta)]. \tag{31}$$

Similarly, to approximate the intractable second term, we must use

$$\max_d \alpha_R \mathbb{E}_{d^{\mathcal{D}}}[\zeta \cdot R]$$

$$- \alpha_Q \mathbb{E}_{\substack{(s,a,r,s')\sim d^{\mathcal{D}} \\ a'\sim\pi(s')}}\left[f_1^*\left(\frac{\zeta(s',a') - (1-\gamma)\mu_0(s')\pi(a'|s') - \gamma\zeta(s,a)}{\alpha_Q d^{\mathcal{D}}}\right)\right] - \alpha_\zeta \mathbb{E}_{d^{\mathcal{D}}}[f_2(\zeta)],$$

which will introduce bias.

# E  Undiscounted MDP

When $\gamma = 1$, the value of a policy is defined as the average per-step reward:

$$\rho(\pi) := \lim_{t_{\text{stop}}\to\infty} \mathbb{E}\left[\frac{1}{t_{\text{stop}}}\sum_{t=0}^{t_{\text{stop}}} R(s_t, a_t)\,\middle|\, s_0 \sim \mu_0, \forall t, a_t \sim \pi(s_t), s_{t+1} \sim T(s_t, a_t)\right]. \tag{32}$$

The following theorem presents a formulation of $\rho(\pi)$ in the undiscounted case:

**Theorem 3.** *Given a policy $\pi$ and a discounting factor $\gamma = 1$, the value $\rho(\pi)$ defined in (32) can be expressed by the following d-LP:*

$$\max_{d:S\times A\to\mathbb{R}} \mathbb{E}_d[R(s,a)], \quad \text{s.t.,} \quad d(s,a) = \mathcal{P}_*^{\pi}d(s,a) \text{ and } \sum_{s,a} d(s,a) = 1. \tag{33}$$

*The corresponding primal LP under the undiscounted case is*

$$\min_{Q:S\times A\to\mathbb{R}} \lambda, \quad \text{s.t.,} \quad Q(s,a) = R(s,a) + \mathcal{P}^{\pi}Q(s,a) - \lambda. \tag{34}$$

*Proof.* With the additional constraint $\sum_{s,a} d(s,a) = 1$ in (33), the Markov chain induced by $\pi$ is ergodic with a unique stationary distribution $d^* = d^{\pi}$, so the dual objective is still $\rho(\pi)$ by definition. Unlike in the discounted case, any optimal $Q^*$ with a constant offset would satisfy (34), so the optimal solution to (34) is independent of $Q$. $\qquad\square$

# F  Experiment Details

## F.1  OPE tasks

For all tasks, We use $\gamma = 0.99$ in all experiments except for the ablation study of normalization constraint where $\gamma = 0.995$ and $\gamma = 1$ are also evaluated. We collect 400 trajectories for each of the tasks, and the trajectory length for Grid, Reacher, and Cartpole are 100, 200, and 250 respectively for $\gamma < 1$, or 1000 for $\gamma = 1$. We run each experiment on 10 seeds and plot the mean and standard deviation of the results.

**Grid.** We use a $10 \times 10$ grid environment where an agent can move left/right/up/down. The observations are the $x, y$ coordinates of this agent's location. The reward of each step is defined as $\exp(-0.2|x - 9| - 0.2|y - 9|)$. The target policy is taken to be the optimal policy for this task (i.e., moving all the way right then all the way down) plus $0.1$ weight on uniform exploration. The behavior policies $\pi_1$ and $\pi_2$ are taken to be the optimal policy plus $0.7$ and $0.3$ weights on uniform exploration respectively.

**Reacher.** We train a deterministic policy on the Reacher task from OpenAI Gym [3] until convergence, and define the target policy to be a Gaussian with the pre-trained policy as the mean and $0.1$ as the standard deviation. The behavior policies $\pi_1$ and $\pi_2$ have the same mean as the target policy but with $0.4$ and $0.2$ standard deviation respectively.

**Cartpole.** We modify the Cartpole task from OpenAI Gym [3] to infinite horizon by changing the reward to $-1$ if the original task returns termination and $1$ otherwise. We train a deterministic policy on this task until convergence, and define the target policy to be the pre-trained policy (weight $0.7$) plus uniform random exploration (weight $0.3$). The behavior policies $\pi_1$ and $\pi_2$ are taken to be the pre-trained policy (weight $0.55$ and $0.65$) plus uniform random exploration (weight $0.45$ and $0.35$) respectively.

### F.2 Linear Parametrization Details

To test estimation robustness to scaling and shifting of MDP rewards under linear parametrization, we first determine the estimation upper bound by parametrizing the primal variable as a linear function of the one-hot encoding of the state-action input. Similarly, to determine the lower bound, we parametrize the dual variable as a linear function of the input. These linear parametrizations are implemented using feed-forward networks with two hidden-layers of $64$ neurons each and without non-linear activations. Only the output layer is trained using gradient descent; the rest layers are randomly initialized and fixed. The true estimates where both primal and dual variables are linear functions are verified to be between the lower and upper bounds.

### F.3 Neural Network Details

For the neural network parametrization, we use feed-forward networks with two hidden-layers of $64$ neurons each and ReLU as the activation function. The networks are trained using the Adam optimizer ($\beta_1 = 0.99$, $\beta_2 = 0.999$) with batch size $2048$. The learning rate of each task and configuration is found via hyperparameter search, and is determined to be $0.00003$ for all configurations on Grid, $0.0001$ for all configurations on Reacher, and $0.0001$ and $0.00003$ for dual and primal regularization on Cartpole respectively.

## G  Additional Results

### G.1  Comparison to unregularized Lagrangian

We compare the best performing DICE estimator discovered in our unified framework to directly solving the Lagrangian without any regularization or redundant constraints, *i.e.*, DR-MWQL as primal, MWL as dual, and their combination [28]. Results are shown in Figure 5. We see that the BestDICE estimator outperforms the original primal, dual and Lagrangian both in terms of training stability and final estimation. This demonstrates that regularization and redundant constraints are crucial for optimization, justifying our motivation.

### G.2  Primal Estimates with Target Networks

We use target networks with double $Q$-learning [11] to improve the training stability of primal variables, and notice performance improvements in primal estimates on the Reacher task in particular. However, the primal estimates are still sensitive to scaling and shifting of MDP rewards, as shown in Figure 6.

Figure 5: Primal (orange), dual (green), and Lagrangian (gray) estimates by solving the original Lagrangian without any regularization or redundant constraints, in comparison with the best DICE estimates (blue).

Figure 6: Primal (red) and Lagrangian (orange) estimates under the neural network parametrization with target networks to stabilize training when rewards are transformed during training. Estimations are transformed back and plotted on the original scale. Despite the performance improvements on Reacher compared to Figure 2, the primal and Lagrangian estimates are still sensitive to the reward values.

## G.3 Additional Regularization Comparison

In addition to the two behavior policies in the main text (i.e., $\pi_1$ and $\pi_2$), we show the effect of regularization using data collected from a third behavior policy ($\pi_3$). Similar conclusions from the main text still hold (i.e., dual regularizer is generally better; primal regularizer with reward results in biased estimates) as shown in Figure 7.

Figure 7: Dual estimates when $\alpha_R = 0$ (dotted line) and $\alpha_R = 1$ (solid line) on data collected from a third behavior policy ($\pi_3$). Regularizing the dual variable (blue) is better than or similar to regularizing the primal variable (orange).

## G.4 Additional Ablation Study

We also conduct additional ablation study on data collected from a third behavior policy ($\pi_3$). Results are shown in Figure 8. Again we see that the positivity constraint improves training stability as well as final estimates, and unconstrained primal form is more stable but can lead to biased estimates.

Figure 8: Apply positive constraint and unconstrained primal form on data collected from a third behavior policy ($\pi_3$). Positivity constraint (row 1) improves training stability. The unconstrained primal problem (row 2) is more stable but leads to biased estimates.