[Reviews · NeurIPS 2020]

Review 1

Summary and Contributions: The paper provides an analysis of the DIstribution Correction Estimation (DICE) estimators. The authors show that the previous DICE approaches can be formulated as regularized Lagrangian of the same LP. Several forms of regularizations are proposed, some of which reduce to previous DICE approaches (like DualDICE, GenDICE, BestDICE). An experimental evaluation comparing the effects of the regularizations, including new combinations, is provided.

Strengths: The main point of strength of the paper is the attempt to put together the different DICE approaches and provide a unified formulation. This formulation allows exploring new forms of regularizations that might turn out to perform better in practice. An extensive experimental evaluation of the different combinations is presented.

Weaknesses: - Theorem 1: I have no doubts about the validity of the theorem, but I have some concerns about its usefulness. As the authors state also in the proof, for both optimization problems d-LP and Q-LP the constraint (which are the recursive definition of the \gamma-discounted distribution and the Bellman equation respectively) determines a unique feasible solution. Thus, I wonder why stating the objective function? - Experiments: it is not reported the number of runs employed to generate the plots of the experiments. Moreover, the plots report some variance areas but it is not clear whether they correspond to some confidence intervals or simply standard deviation. ***Minor*** - Footnote 1 and line 54: Do you focus on \gamma \in (0,1] or \gamma \in [0,1)? - Line 98: \mathcal{P} is not defined - Section 3.3: maybe a table would present the reduction to the known cases more clearly - Figures: the labels on the axis and the legends are too small

Correctness: All claims are proved in the appendix.

Clarity: The paper is written in good English and reads well.

Relation to Prior Work: The related works are adequately discussed in Section 5, with particular reference to off-policy evaluation based on importance sampling and DICE approaches. Moreover, a paragraph about convex duality applied to RL is reported.

Reproducibility: Yes

Additional Feedback: I think that the paper does a fairly good job in generalizing the DICE approaches, providing a unified view and an extensive experimental evaluation. I didn't find any fault in the paper, but I am not sure whether it is a sufficiently significant contribution for NeurIPS. For these reasons, I am inclined towards weak acceptance. ***Post rebuttal*** I thank the authors for the feedback. I read it together with the other reviews. I appreciate the clarification about the contributions of the paper, but I still think that this work represents an incremental contribution. Moreover, my doubts about the usefulness of Theorem 1, beyond its usage in deriving the dual, remain. I will keep my original evaluation.


Review 2

Summary and Contributions: ======================= I think the paper is a good one and will keep my original high score. I want to note however, that following both the authors' feedback and discussion among reviewers, I feel the other reviewers raised valid concerns regarding the fact that the contribution of the paper is not apparent to anyone who is not closely following the DICE line of work. I therefore encourage the authors to try and make that contribution more accessible to a wider audience, regardless of what the acceptance decision will be. Good luck! ======================= The paper presents a formulation which unifies multiple recent off-policy evaluation methods. By formulating the difficult optimization inherent to all of these problem as a regularized Lagrangian of a linear programing problem, the authors manage to give a unified view of the problem, and propose a new optimization method which performs well in practice, as demonstrated empirically.

Strengths: The DICE methods for OPE which are the subject of the paper have recently gained much popularity, but they require very difficult optimization, and the authors novel approach to the problem will be impactful. Furthermore, this line of research (DICE) has been very fruitful over the past two years, and many papers have tried different approaches to the optimization problem. Thus, the analysis unifying the different approaches is great in that it grounds past and future research in a sub-field which was starting to feel like a niche composed of bunch of optimization hacks.

Weaknesses: The one weakness of the paper is that it is extremely specialized. I think it would be of interest only to a small subgroup of researchers who are very detail oriented to the the DICE approach and the difficulties of its optimization. Furthermore, the paper is by no means self contained and requires deep understanding of optimization methods. Having said that, I would like to defend the paper by commenting that I think this level of specialization is required, and the "stationary state distribution" approach to OPE is one of the more promising directions in OPE which is currently being pursued, and so solving the optimization problem can have huge impact on OPE and by extension applications of RL to the real world.

Correctness: The paper is correct as far as I can tell. However, I don't have the background to follow the fine details of some of the optimization tricks.

Clarity: The paper is written clearly, but assumes high familiarity with optimization methods.

Relation to Prior Work: The paper does a good job of placing itself in the context of previous works.

Reproducibility: Yes

Additional Feedback: The authors show how DualDICE and GenDICE can be recovered by their formulation. However, since so much of these methods come down to optimization stability, it would be interesting to see an empirical comparison of the results for the direct implementation of these methods and the "recovered" implementation from the regularized lagrangian.


Review 3

Summary and Contributions: This paper proposes an off-policy evaluation method based on offline historical trajectory data. The method in this paper assumes that the data is generated from an offline distribution, and then uses the duel Linear Programming method to fit the distribution of the learning target policy, and then adopt the Regularizations and Redundant Constraints method to stabilize the process of objective function optimization. The experimental part verifies the effectiveness of the method. It is compared with True value on the three data sets of Grid, Reacher and Cartpole to verify the effect of policy evaluation. In addition, the author verifies that Regularizations are effective for training stability through different parameters.

Strengths: 1. The problem in this article to solve is crucial in the field of reinforcement learning. Most of the problems to be solved in the real world do not have an environment, and it is often unrealistic to interact with the environment to get the policy result of a fixed policy, because it will bring a lot of interaction costs. Therefore, it is very important to evaluate strategies based on historical trajectory data only. This paper proposes an effective method to solve this offline data policy evaluation problem. 2. This article introduces Regularizations and Redundant Constraints in the optimization process to make the training process stable, and the experimental results also prove the effectiveness of this method. I think it is very crucial in offline policy evaluation, because historical trajectory data is generated from historical distribution, so if regularizations are not added, it is very likely to overfit the data distribution. Therefore, when the offline evaluation data distribution changes greatly, it will affect the effect of model learning.

Weaknesses: 1. The biggest problem with this article is that innovation and contribution are not enough. For example, most of the content and formulas of the Section 2, off-policy evaluation are basically the same as those in the DualDICE paper. Mainly the third part introduces the method of this article. From the objective function and the basic problem to be solved, I think the objective function the author try to solve is identical to that proposed in DualDICE paper . The only difference is that the author uses the augmented Lagrangian method to optimize the problem and get the optimization result. Although the author stated a difference from DualDICE and other methods in Chapter 3.3, in general, this article is indeed insufficient in terms of innovation. 2. The second problem of this article is the experimental part. Most of the author's comparison methods are to select different distribution and parameters, comparing the method of this article and the error of the policy accumulate reward. I suggest the author to compare with other state-of-the-art methods. Because there is no evidence to conclude that this article performs better than other state-of-the-art methods, when only comparing with the method of this article. I browsed all the supplementary materials, and found no experimental effects of other methods.

Correctness: Yes, empirical methodology part seems good for me.

Clarity: Yes, the paper is well organized.

Relation to Prior Work: This article introduces the difference between the proposed method and DualDICE in some corner cases in Chapter 3.3. But it is not compared with other methods in the experiment.

Reproducibility: Yes

Additional Feedback:


Review 4

Summary and Contributions: This paper tries to unify the recent minimax approaches for off-policy evaluation using Lagrangian. The main contribution is the general form of a minimax problem derived using LP optimization in both primal and dual, whose Lagrangian can be reduced to different OPE methods as special cases. These methods are all (partially) focused on estimating a state action distribution correction weight for the policy evaluation. Empirically, this paper compares different methods the general form can derive and investigate the tradeoff between optimization stability and estimation bias. The main observation is that the dual estimators usually outperforms the other estimators. ============== Thanks for the author response. I changed the score to 6 because I get a better understanding of the paper after reading the rebuttal. I do still have the confusion of how to interpret thm1. Why we use a constrained reward maximization (for the dual) to do OPE? I did not get a very convincing interpretation. I agree the technical content is sound. But I am not sure how novel and interesting the insight is. Also, I feel the "new" algorithm proposed is not that new. The useful insights for locating a better variants (here seems to be using positivity and \lambda) is not clear to me and not novel. I think the paper would be much stronger if there is an interesting new direction for developing OPE method.

Strengths: This paper’s contribution is mainly in the unification it provides to various OPE methods in terms of derivation and the illustration of the connection and difference between them, which is generally a nice contribution to the field. The empirical study is extensive in terms of the variants of the method it covers. The writing is clear.

Weaknesses: The presentation of the experimental results can be probably improved. The layout of the small figures does not convey the information effectively. Theorem 1 needs more explanation on why it is a valid start point for OPE problems. I would expect a proposal of a new method, building on unification and comparison of various methods.

Correctness: In terms of correctness, I have the following questions: Theorem 1 starts with a formulation that either finds the visitation distribution that maximizes the expected reward under constraints given by target policy or finds the value function that minimizes the expected value function on target policy under constraints. Even though it seems to be correct, I was wondering why these are reasonable start point for an OPE problem. From previous literature, their starting point may be the unknown function class of weight estimation so referring to the worst-case possible case. But in this case, the intuition is not clear to me. The derivation of (8) comes from first the Lagrangian and then change of variables of the weight. However, from the primal LP, why the Lagrangian multipliers (dual parameters) are the visitation distribution. Is d(s, a) here just the weight or it still has the meaning of a visitation distribution of target policy?

Clarity: This paper is generally well-written, especially in the main theoretical content. However, given my previous questions, more remarks and explanations can be given to improve the clarity. The experimental section presents an extensive set of results comparing the estimation bias and the training stability. However, the main message is not clear. My suggestion is to only show the converged methods and also only show the interval after convergence, with notes for the unconverged ones. This way, the curve would look less noisy. Using tables to summarize results could be another alternative to present clean results.

Relation to Prior Work: The related work is surveyed extensively.

Reproducibility: Yes

Additional Feedback: This paper generally provides good insights for OPE methods. However, after the unification and extensive experiments, I am expecting the proposal of a method that can overcome some of the previous problems and be accurate and stable at the same time. For example, is there a meta-algorithm that chooses the estimator and the parameter for a given problem? So it seems there is still one step not completed.

[Author Response · NeurIPS 2020]

We thank the reviewers for their helpful comments and address the individual comments below.

**Reviewer #1. Sufficiency in contribution.** 1) BestDICE *is* novel and outperforms all existing DICE estimators, as
shown in Fig. 1 below (variants of this figure appeared in the original submission where legends are regularization
configurations rather than estimator names). 2) We also derived a comprehensive bias analysis for an expanded family
of DICE estimators (Theorem 2 in the main text and Table 1 in the appendix), whereas previous DICE papers only show
a particular algorithm being (almost accidentally) unbiased. Theorem 2 and Table 1 present a foundation for future
distribution-based OPE research by providing theoretical guarantees for the choices of estimators and regularizers.
**Objectives in Theorem 1.** The objectives connect the LP solution ($Q^\pi$ or $d^\pi$) to the policy value $\rho(\pi)$, which is what
OPE ultimately cares about.

**Reviewer #2. Specialization.** While this work focuses entirely on OPE, we believe it is also a strength, given the
widely recognized importance of the OPE problem and the current proliferation of proposed algorithms. Indeed,
our regularized Lagrangian formulation provides a novel unification, which shows that many of these algorithms are
actually obtained simply by choosing alternative regularizations. **Direct and recovered implementation.** The current
ecosystem of open-sourced DICE implementations is unfortunately fragmented and incomplete. A key empirical
contribution of this work is indeed to provide a unified implementation of all DICE algorithms, where we have also
verified that our implementation reproduces the results reported in previous DICE papers. (Our open-sourced code has
already been released, but we need to suppress any links to preserve review anonymity.)

**Reviewer #3.** There are several misunderstandings and inaccuracies in this review. **1)** *"This paper proposes an*
*off-policy evaluation method based on offline historical trajectory data."* — The paper's goal is to provide a unified
view of DICE estimators, covering both existing and new methods, and understanding the impact of various algorithmic
choices. **2)** *"The experimental part verifies the effectiveness of the method."* — The experiments are not to verify
any method, but to analyze the impact of regularization on solution biases and optimization stability. **3)** *"The biggest*
*problem with this article is that innovation and contribution are not enough. For example, most of the content and*
*formulas of the Section 2, off-policy evaluation are basically the same as those in the DualDICE paper."* — Kindly
observe that Section 2 is the background section intended to set up the problem formulation and notation, and has
nothing to do with the work's novelty. **4)** *" the objective function ... is identical to ... DualDICE"* — This assertion
is false, since objective we consider contains $R(s,a)$ and $f(Q)$, which never appeared in the DualDICE objective.
DualDICE and other recent algorithms (Sec. 3.3) can be seen as particular ablations (see, e.g., line 192). **5)** *"If*
*regularizations are not added, it is very likely to overfit the data distribution."* — In this context, regularization was
introduced to the Lagrangian to stabilize optimization (line 136), not to address overfitting. **6)** *"The only difference is*
*that the author uses the augmented Lagrangian method"* — We are *not* using an augmented Lagrangian method, which
would lead to a double sampling problem as explained in Sec 3.2. We have had to therefore develop to an alternative
approach. **7)** *"Only comparing with the method of this article"* and *"not comparing with other state-of-the-art methods"*
— The recent DICE estimators are considered state-of-the-art in OPE, and we compared to all such methods recoverable
from the regularized Lagrangian. It is unfortunate no particular work was pointed out to support such an assertion.

**Reviewer #4. Theorem 1 as OPE starting point.** The constraints in the theorem characterize the dual and primal
quantities ($d^\pi$ and $Q^\pi$), which can be used to estimate policy value, either alone or combined (lines 171-173, with
a change-of-variable $\zeta = d^\pi/d^\mathcal{D}$). It is thus a natural starting point for OPE, which we will make explicit in the
final version. The variables $d(s,a)$ play the roles of both Lagrangian multipliers and the visitation distribution: The
Lagrangian of the primal LP is $\mathcal{L} = (1-\gamma)\mu_0^\top Q + d^\top(R + \gamma PQ - Q)$, with multipliers $d$. By taking the gradient of
$\mathcal{L}$ with respect to $Q$ and setting the gradient to 0, we get $d = (1-\gamma)\mu_0 + \gamma P^\top d$, which is exactly what the stationary
state-action visitation satisfies (Eq. (4) in the paper). **Proposal of a new method.** We did propose a new method,
*BestDICE*, which outperforms others (Fig 1 below). We consider it possible to develop new meta-algorithms for
model selection that can work better than BestDICE. **Experiment presentation.** We present the estimates produced
during training to highlight the optimization behavior, as our major empirical contributions is to systematically apply
regularizations to solve the challenging minimax optimization problem present in previous DICE algorithms.

Figure 1: Comparison of BestDICE to other state-of-the-art OPE methods. Note that variants of this figure appeared in
the original submission under different legends (e.g., rather than using GenDICE as the legend, we used "Dual est. +
Primal reg. + Positivity + Normalization").

[Meta-Review · NeurIPS 2020]

Reviewers generally agreed that this paper makes a good contribution in unifying the DICE estimators that provides useful insights.